# Effects of *Pueraria candollei* var *mirifica* (Airy Shaw and Suvat.) Niyomdham on Ovariectomy-Induced Cognitive Impairment and Oxidative Stress in the Mouse Brain

**DOI:** 10.3390/molecules26113442

**Published:** 2021-06-05

**Authors:** Yaowared Chulikhit, Wichitsak Sukhano, Supawadee Daodee, Waraporn Putalun, Rakvajee Wongpradit, Charinya Khamphukdee, Kaoru Umehara, Hiroshi Noguchi, Kinzo Matsumoto, Orawan Monthakantirat

**Affiliations:** 1Division of Pharmaceutical Chemistry, Faculty of Pharmaceutical Sciences, Khon Kaen University, Khon Kaen 40002, Thailand; yaosum@kku.ac.th (Y.C.); Berzerkwiz@gmail.com (W.S.); csupawad@kku.ac.th (S.D.); rakvajeewpd@kkumail.com (R.W.); 2Division of Pharmacognosy and Toxicology, Faculty of Pharmaceutical Sciences, Khon Kaen University, Khon Kaen 40002, Thailand; waraporn@kku.ac.th (W.P.); charkh@kku.ac.th (C.K.); 3Department of Pharmacognosy, School of Pharmaceutical Sciences, University of Shizuoka, Yada 52-1, Shi-zuoka-shi, Shizuoka 422-8526, Japan; kaoru.umehara@hamayaku.ac.jp (K.U.); noguchi@u-shizuoka-ken.ac.jp (H.N.); 4Faculty of Pharmaceutical Sciences, Yokohama University of Pharmacy, Yokohama, Kanagawa 245-0066, Japan; 5Department of Pharmacognosy, Nihon Pharmaceutical University, Saitama 362-0806, Japan; 6Division of Medicinal Pharmacology, Institute of Natural Medicine, University of Toyama, 2630 Sugitani, Toyama 930-0194, Japan; mkinzo@inm.u-toyama.ac.jp

**Keywords:** *Pueraria candollei* var. *mirifica*, ovariectomy, cognitive dysfunction, oxidative damage, neuroinflammation

## Abstract

The effects of the phytoestrogen-enriched plant *Pueraria mirifica* (PM) extract on ovari-ectomy (OVX)-induced cognitive impairment and hippocampal oxidative stress in mice were investigated. Daily treatment with PM and 17β-estradiol (E2) significantly elevated cognitive behavior as evaluated by using the Y maze test, the novel object recognition test (NORT), and the Morris water maze test (MWM), attenuated atrophic changes in the uterus and decreased serum 17β-estradiol levels. The treatments significantly ameliorated ovariectomy-induced oxidative stress in the hippocampus and serum by a decrease in malondialdehyde (MDA), an enhancement of superoxide dismutase, and catalase activity, including significantly down-regulated expression of IL-1β, IL-6 and TNF-α proinflammatory cytokines, while up-regulating expression of PI3K. The present results suggest that PM extract suppresses oxidative brain damage and dysfunctions in the hippocampal antioxidant system, including the neuroinflammatory system in OVX animals, thereby preventing OVX-induced cognitive impairment. The present results indicate that PM exerts beneficial effects on cognitive deficits for which menopause/ovariectomy have been implicated as risk factors.

## 1. Introduction

Accumulated evidence over the last 30 years has demonstrated that the estrogen hormone modulates cognitive function in female rodents and primates, including humans [1]. Modulation by estrogens initially targets the ovary and uterus through which estrogens play a role in the sexual differentiation of various brain functions, including the regulation of reproduction and some cognitive functions. With aging, circulating estrogen levels markedly decrease and appear to contribute to age-related decline in learning and memory function in females [2]. These estrogen-dependent effects on cognition function throughout the lifespan have been attributed to classic genomic mechanisms, including processes such as hormone binding to specific receptors, alterations in gene transcription, and the initiation of organ-specific effects in target areas in the brain and peripheral organs [3]. Moreover, increasing evidence suggests that gonadal hormones act as neurosteroids that are produced in the brain and, thus, rapidly alter cognition and other neural functions [4]. Therefore, the decrease induced in estrogen levels by menopause or ovariectomy may not only increase the incidence of inflammatory pathologies, but also decrease neurogenesis and neuronal plasticity [5].

Clinical studies have demonstrated that the incidence rates of neurodegenerative diseases, including Alzheimer’s disease (AD), are high in post-menopausal women and this increase has been attributed to the decrease in estrogens levels caused by menopause [6]. A putative mechanism underlying the epidemiological relationship between the decrease in estrogen and the incidence of AD has also been suggested by several lines of study [1]. For example, estrogen has been shown to not only enhance the non-amyloidogenic cleavage of APP by increasing the expression of α-secretase, but also facilitates the clearance of Aβ by promoting its phagocytosis by microglia [7]. Moreover, estrogen appears to have the ability to reduce tau aggregates and stimulate the expression of choline acetyltransferase (ChAT) in the basal forebrain, thereby slowing the cognitive decline caused by AD [8]. However, other factors may also contribute to the pathophysiology of AD, such as mitochondrial dysfunction, oxidative stress, neurotransmitter failure, and inflammation [9,10]. The brain is a postmitotic organ that is susceptible to oxidative damage because of the high utilization of inspired oxygen, a large amount of easily oxidizable polyunsaturated fatty acids, an abundance of redox-oxidative transition metal ions, and relative deficiency in the antioxidant defense system [11]. Evidence suggests that the disruption of CNS areas results in various mental illnesses and cognitive dysfunction in menopausal women and an imbalance in the CNS redox state may influence these pathologies [12]. Hormone replacement therapy (HRT) is the main choice to prevent or attenuate menopause-associated alterations. However, complementary studies have reported a number of adverse effects associated with long-term HRT, such as an increased risk of breast cancer, endometrial cancer, stroke, and thromboembolic events [13]. Thus, an alternative phytoestrogen may be more beneficial than conventional HRT with a poor safety profile or side effects [14]. *Pueraria candollei* var. mirifica (Airy Shaw and Suvat). Niyomdham is a Thai herbal plant known as “Kwao Krua Kao” in Thai (commonly termed *P. mirifica*, PM). The traditional uses of PM were initially described in a book written by Suntara in 1931 as the “fountain of youth” for the elderly. It was used to smoothen skin, promote hair growth, improve cognitive function, increase blood circulation, alleviate sleep disorders, and increase energy and vigor [15,16]. The tuberous root of this plant has been used by local communities in Thailand for its rejuvenating qualities in menopausal women. Thai people also used it to recover black hair, promote an appetite, and increase their longevity [17]. The ordinary dosage of PM for women is a peppercorn-sized piece, which is equivalent to approximately 250 mg/kg body weight, taken once daily at night [18]. The ethnobotanical application of PM in folk medicine implies the recognition of its multiple effects in the elderly without knowledge of hormones in that era. This indicates that it contains some active compounds that exert similar effects to the female hormone [16] *P. mirifica* extract and its products have been attracting increasing attention worldwide because of their pharmacological activities against estrogen deprivation-related disorders [19]. At least 13 compounds have been isolated from the tuberous roots of this plant as phytoestrogens. They include puerarin, puerarin-6’-monoacetate, daidzin, genistin, daidzein, genistein, mirificin, kwakhurin, kwakhurin hydrate, coumestrol, miroestrol, isomiroestrol, and deoxymiroestrol [20]. The estrogenic activities of PM extracts have been tested widely in cell cultures, animals, and humans [10,16,18,21]. We previously reported that miroestrol-treated OVX mice exhibited improved cognitive function and neuroprotective effects in the hippocampal region [10]. In addition, PM extract and miroestrol were shown to improve the oxidative status of the livers and uteri of OVX mice and prevented bone loss by increasing the ratio of osteoprotegerin (OPG) to a receptor activator of nuclear factor kappa B ligand (RANKL) (osteoformation/osteoresorption) in the livers of ovariectomized [22]. In clinical trials (phases I, II and III), PM root powder (50–100 mg daily) improved estrogen depletion-induced hot flushes, mood instability, insomnia, muscle pain, headaches, nervousness, and urinary incontinence [23]. Therefore, the main objectives of the present study were to investigate the effects of PM ex-tract ovariectomy-induced cognitive dysfunction and oxidative stress in mouse serum and the hippocampus and compare the effects of PM extract with those of 17β-estradiol.

## 2. Results

### 2.1. Effects of PM Extract and Estrogen on OVX-Induced Cognitive Impairments

In order to establish whether PM extract modulates OVX-induced cognitive impairments, the spatial and non-spatial memory performances of OVX mice were elucidated by the Y-maze test and novel ORT, respectively. The vehicle-treated OVX group exhibited significantly less spontaneous alternations than the sham-operated group, indicating the impairment of spatial working memory caused by estrogen deprivation 17β-Estradiol and PM-treated OVX groups exhibited significantly better spontaneous alternation performance in the test than the vehicle-treated OVX group (Figure 1A) (for detailed statistical analysis, see Appendix A).

The novel ORT also revealed OVX-induced cognitive deficits in mice. In the sample phase trial in which two identical objects were used, none of the animal groups exhibited significant differences in the time spent exploring each object (data not shown). However, in the test phase trial, the sham-operated group spent a significantly longer time exploring the novel object than the familiar one, while vehicle-treated OVX mice failed to discriminate these two objects, indicating the OVX-induced impairment of non-spatial working memory. On the other hand, OVX mice that received 17β-estradiol and PM extract for 8 weeks exhibited significantly improved discrimination performance in the test phase trial (Figure 1B) (for detailed statistical analysis, see Appendix A).

The effects of PM extract and 17β-estradiol on the spatial reference memory of OVX mice were also elucidated using the MWM test. All groups acquired the platform location, as supported by decreased escape latency, indicating some degree of learning ability. Statistical analyses revealed that vehicle-treated control OVX mice had significantly weaker abilities to learn the location of hidden platform and retrieve the platform location in the training phase and probe test, respectively, than sham-operated mice. This result indicated OVX-induced reference memory impairment. The repeated administration of 17β-estradiol and PM extract significantly ameliorated the acquisition and retrieval performance of reference memory impaired by estrogen deprivation (Figure 1C,D) (for detailed statistical analysis, see Appendix A).

### 2.2. Changes in Uterus Weight and Volume and Serum 17β-Estradiol Levels after the PM Treatment

Uterus weights and volumes as well as serum 17β-estradiol levels were significantly lower in OVX mice than in sham-operated mice (Table 1). HRT for 8 weeks with 17β-estradiol in OVX mice significantly reversed OVX-induced atrophy of the uterus and decreased serum 17β-estradiol levels. Similarly, the daily treatment of OVX animals with PM extract significantly and dose-dependently attenuated the atrophic effects of OVX on the uterus. The 17β-estradiol treatment significantly increased serum 17β-estradiol levels in OVX animals, whereas the treatment with PM at a dose of 25 mg/kg/day slightly reversed the OVX-induced decrease in serum 17β-estradiol levels (for detailed statistical analysis, see Appendix A).

### 2.3. Effects of PM Extract and Estrogen on Oxidative Damage and Antioxidant Enzyme Activities in the Hippocampus and Serum of OVX Mice

We investigated the effects of PM extract and 17β-estradiol treatments on OVX-induced oxidative damage (Figure 2A,B) and reductions in the activities of SOD (Figure 2C,D) and CAT (Figure 2E,F), antioxidant enzymes, in the hippocampus and serum. The accumulation of the aldehyde product of lipid peroxidation, a marker of oxidative damage, in the hippocampus and serum was markedly greater in OVX mice than in sham control mice. In addition, OVX mice also showed significant reductions in the activities of SOD and CAT. HRT with 17β-estradiol (1 ug/kg/day, i.p.) and daily supplementation with PM extract (25 mg/kg/day, p.o.) in OVX mice completely prevented OVX-induced lipid peroxidation and recovered SOD and CAT activities in the hippocampus and serum (for detailed statistical analysis, see Appendix A).

### 2.4. Effects of PM Extract and Estrogen on Proinflammatory Cytokines and Estrogen-Mediated Gene in Hippocampus

Proinflammatory cytokines, interleukin 1β (IL-1β) (Figure 3A), interleukin 6 (IL-6) (Figure 3B) and tumor necrosis factor α (TNF-α) (Figure 3C) and estrogen-mediated gene, PI3K mRNA (Figure 3D) in hippocampus were evaluated by quantitative real-time PCR (QPCR) analysis. All mRNA of proinflammatory cytokines were significantly increased whereas PI3K mRNA was decreased in hippocampus of the OVX mice when compared with sham control mice. The daily administration of OVX animals with PM extract significantly down-regulated expression of three proinflammatory cytokines while up-regulated expression of PI3K as well as administration of 17β-estradiol (for detailed statistical analysis, see Appendix A).

## 3. Discussion

Post-menopausal women in an estrogen-depleted state are at risk of neurodegenerative disease with a decline in cognitive brain function [3,6]. Previous studies demonstrated that HRT is beneficial not only for various menopausal symptoms, but also reduces the risk of AD after climacterium [12,24]. Based on these findings, the present study aimed to investigate using OVX animals whether PM is applicable as an effective alternative to HRT for reducing the risk of cognitive deficits after menopause or ovariectomy [10]. The results obtained demonstrated that the daily administration of PM extract attenuated the deterioration of spatial and non-spatial cognitive functions in an animal model of menopause/ovariectomy and also that the ameliorative effects of the extract on cognitive dysfunction were at least partly mediated by the suppression of oxidative hippocampal damage attributable to the decreased activities of anti-oxidative enzymes.

The mouse model of OVX used in the present study showed clearly atrophic changes in the uterus, a decrease in the circulating level of E2, and cognitive deficits in a manner that may be reversed by HRT with 17β-estradiol. These results are consistent with previous findings from ours and other groups [8,10,14,16], indicating that the animal model prepared was appropriate for mimicking the estrogen-depleted state of females and elucidating the pharmacological activities of drugs, including natural herbal medicines such as PM.

The present behavioral study clearly demonstrated that the daily administration of PM extract as well as 17β-estradiol dose-dependently ameliorated not only the impairment in hippocampus-dependent spatial working memory (the Y-maze test) and reference memory (the MWM test), but also deficits in hippocampus-independent non-spatial working memory (the novel ORT) in OVX animals. These results indicate that PM extract exerts protective effects against cognitive dysfunction attributable to menopause/estrogen depletion, an important risk factor for dementia including AD [2]. Moreover, since PM extract at a daily dose of 25 mg/kg exerted anti-dementia effects that were almost equivalent to the effects elicited by 17β-estradiol at 1 μg/kg/day, the present study provides a rational reason for the traditional use of this plant for the treatment of cognitive dysfunction in Thailand.

The present HRT with 17β-estradiol significantly ameliorated all estrogenic parameters; i.e., uterus weight and volume as well as serum E2 levels in OVX animals [14]. On the other hand, PM extract exerted similar effects to 17β-estradiol in OVX animals, except for the effect on serum E2 levels. These results suggest that PM extract contains a constituent(s) with potential to directly stimulate the receptor responsible for endogenous E2, the estrogen receptor-mediated cellular signaling mechanism, or both [25]. The present HPLC analysis of PM extract identified various phytoestrogens that included the chromene-enriched part containing miroestrol, deoxymiroestrol, and isomiroestrol [26]. Among these chemical constituents, potent estrogenic compounds belong to chromenes. Moreover, deoxymiroestrol reportedly exhibited 6-, 300-, 400-, and 3000-fold stronger estrogenic activity than miroestrol, coumestrol, genistein, and daidzein, respectively, as assayed using a MCF7 human breast cancer cell system [27]. PM extract at doses of 2.5 and 25 mg/kg/day was equivalent to 0.050 and 0.505 mg of miroestrol and 0.025 and 0.250 mg deoxymiroestrol, respectively. Collectively, the present results indicate that the estrogenic and anti-dementia effects of PM extract are mainly due to the stimulation of estrogen receptors by miroestrol and deoxymiroestrol. Moreover, these biological features of miroestrol and deoxymiroestrol included in the PM extract may be beneficial for AD for a number of reasons. The activation of estrogen receptors has been linked to PI3K and MAPK signaling pathways, which play important roles in neurogenesis, synaptic plasticity, and ERK1/2 signaling. Furthermore, facilitation of the ERK1/2 signaling system via estrogen receptors reportedly leads to a decrease in β-amyloid production [8]. Many studies have reported that chronic neuroinflammation in the brain increases the risk for AD [28]. In this study, we found that lacking E2 (OVX) can also contribute to neuroinflammation by increasing IL-1β, IL-6 and TNF-α via the PI3K/AKT-mediated pathway with decreasing PI3K genes transcription. PM and 17β-estradiol treatment recovered their expression. In order to obtain a better understanding of the mechanisms underlying the anti-dementia effects of PM in OVX animals, we herein investigated the effects of PM and HRT on OVX-induced oxidative damage in the hippocampus and serum by measuring MDA levels as an index of lipid membrane damage and antioxidant enzyme activities in OVX animals. Our results showed that OVX decreased SOD and CAT activities and increased MDA levels in the hippocampus and also in serum in a dose-dependent manner that was reversed by the daily administration of PM and 17β-estradiol. The hippocampus plays an important role in cognitive performance, particularly spatial memory, and is the brain area most susceptible to oxidative stress [29]. Moreover, oxidative damage to the hippocampal area under estrogen deprivation conditions and the prevention of damage by chemicals with antioxidant activity may account for alterations in the cognitive performance of OVX animals [30]. Therefore, a plausible explanation for the anti-dementia effects of PM extract in OVX mice is that chemical constituents of the PM extract protected hippocampal antioxidant systems, including SOD and CAT, from OVX-induced damage, thereby attenuating the cognitive dysfunction induced by oxidative stress under estrogen depletion. This concept appears to be supported by previous findings. Azcoitia et al. [31] found that estrogen exhibited potent preventative activity against neuro-degenerative diseases by scavenging free radicals, activating the antioxidant defense system, limiting mitochondrial protein damage, and improving electron transport chain activity. Moreover, estrogen reportedly reversed neurogenesis in the hippocampus and increased dendritic spine density in the CA1 pyramidal neurons of OVX mice [8]. Furthermore, according to Chershewasart et al. [32] who reported a relationship between antioxidant activity and the major isoflavonoid contents of PM, the antioxidant activity of this plant only correlated with the contents of PM phytoestrogens, such as miroestrol and deoxymiroestrol. We previously demonstrated that the daily administration of 0.5–1 mg/kg miroestrol improved cognitive dysfunction and atrophy of the uterus via antioxidant and estrogenic activities without altering serum E2 levels in OVX animals [10]. The present study revealed that OVX induce neuroinflammation in mice brain, leading to cognitive impairment. PM extract and E2 exhibited together the potential important relevance of considering anti-inflammatory and antioxidant properties. Accordingly, the hippocampus has been identified as a target for estrogen. Moreover, to enhance neuronal damage in the brain, OVX was demonstrated to activate astrocytes and microglia in the hippocampus and increased the release of inflammatory cytokines such as IL 6, IL 1β and TNF α, leading to decreases in cognitive function. The present results showed that IL 6, IL 1β and TNF-α levels were significantly decreased and increased PI3K following PM treatment. Furthermore, supporting the theory that PM may reduce neuroinflammation, studies are required to investigate whether PM provides its effects via astrocytes or microglia.

## 4. Materials and Methods

### 4.1. Plant Materials and Extraction

The tuberous root bark of PM was provided by Dr. Waraporn Putalun, Khon Kaen University and was collected in Ubon Ratchathani, Thailand. A reference specimen (NI-PSKKU 007-010) was deposited at the Herbarium of the Faculty of Pharmaceutical Sciences, Khon Kaen University, Thailand. The preparation of PM was described previously by Monthakantirat et al. [10] The miroestrol-rich fraction was identified and analyzed by a gradient high-performance liquid chromatography (HPLC) method using a LichroCart C18 reverse phase column (5 μm, 25 × 0.4 cm). The oven column was set at 30 °C. The gradient elution program was varied by the proportion of solvent A (1.5% acetic acid in water) and solvent B (acetonitrile) with a flow rate at 1.0 ml/min. The UV detection wavelength was set at 280 nm for obtaining chromatograms. Nine phytoestrogens were detected. Six major isoflavones, i.e., puerarin, daidzin, genistin, daidzein, genistein, and kwakhurin, were successively found with the independent retention time (for detailed retention time and HPLC chromatogram, see Appendix A) at concentrations of 0.54, 0.07, 0.09, 1.72, 0.25, and 3.11 mg/100 mg PM extract, respectively. The chromene derivatives, isoiroestrol, miroestrol, and deoxymiroestrol, were present at concentrations of 0.60, 2.02, and 1.00 mg/100 mg PM, respectively. Phytoestrogen contents were in the order of kwakhurin > miroestrol > daidzein > deoxymiroestrol > isomiroestrol > puerarin > genistein > genistin > daidzin.

### 4.2. Animals

Fifty-five 5-week-old female mice were obtained from the National Laboratory Animal Center (Mahidol University, Nakhon Pathom, Thailand). Mice were housed in paper-bedding cages and given food and water ad libitum. Housing conditions were a light-controlled room with 12-h light/dark cycle lights on: (07:00–19:00) under temperature control (24 ± 1 °C) in the Laboratory Animal Unit of the Faculty of Pharmaceutical Sciences (Khon Kaen University, Khon Kaen, Thailand). All animal research procedures were performed in accordance with the Guiding Principles for the Care and Use of Animals (NIH Publications No. 80–23, revised in 1996). The present study was also conducted in accordance with the Animal Ethics Committee for Use and Care, Khon Kaen University, Khon Kaen, Thailand (Approval No. AEKKU 01/2556).

### 4.3. Surgical Procedure

OVX mice were used to mimic an estrogen-deprived state in the present study. Ovariectomy was conducted under pentobarbital anesthesia (Nembutal^®^: 60 mg/kg; Ceva Sante Animale, France) as previously described [10]. Briefly, animals underwent bilateral ovariectomy via a dorsolateral incision. The exposed ovary and associated oviduct were removed and skin incisions were closed. The sham-operated group underwent the same method without the removal of the ovaries. After a 3-day recovery period, animals were divided into five groups: (1) sham, (2) ovariectomy (OVX), (3) ovariectomy + 1 μg/kg 17β-estradiol (OVX+ E2), (4) ovariectomy + 2.5 mg/kg PM (OVX + PM 2.5), and (5) ovariectomy + 25 mg/kg PM (OVX + PM 25). The sham and OVX control groups were orally administered corn oil. 17β-estradiol was intraperitoneally administered and PM was orally administered once daily for 8 weeks. In order to assess the effects of 17β-estradiol and PM in OVX mice, drugs were administered 1 h before the behavioral test. The experiment framework is summarized in Figure 4.

### 4.4. Behavioral Test

#### 4.4.1. Y-Maze Test

The Y-maze was used to examine the hippocampus-dependent spatial working memory of animals. The Y-maze consisted of three arms: 40 cm long, 18 cm high, 3 cm wide at the bottom, and 12 cm wide at the top, which were positioned at equal angles. The maze floor and walls were constructed from dark opaque polyvinyl plastic. The Y-maze test was conducted 1 h after drug administration. Animals were individually placed on one arm, and the sequence of arm entries was recorded manually over an 8-min period. An actual alternation was defined as entries into all three arms on consecutive choices (i.e., 123, 312, or 231, but not 212). When all fours limbs were within an arm, an animal was judged to have entered it [33]. Percent alternation was calculated according to the following Equation (1):% alternation = [(number of alternations)/(total arm entries − 2)] × 100.(1)

Maze arms were cleaned using 70% ethanol between tasks to remove residual odors.

#### 4.4.2. Novel Object Recognition Test (NORT)

The novel ORT was performed as described previously [10,34]. The apparatus consisted of a square arena (50 × 50 × 40 cm high). The height of the objects was sufficient to prevent the mice from climbing on them. The NORT consisted of three different sessions: habituation, a sample phase trial, and test phase trial. Approximately 24 h before the test, each mouse was individually habituated to the test box, with 10 min exploration in the absence of objects. In the sample phase trial, each mouse was placed into the observation box, in which two identical objects (objects O1 and O2) were placed in two adjacent corners, and allowed to explore for 5 min. Mice were considered to be exploring the object when the head of the animal was facing the object or when the animal was touching or sniffing the object. The exploration time of each object was measured. In the test phase trials performed 30 min after the sample phase trials, one of the two objects was replaced by a novel object. The total time spent exploring each familiar object and the novel object was analyzed. The box arena and objects were cleaned using 70% ethanol between trials to prevent a build-up of olfactory cues. Memory was assessed by measuring the animal’s ability to recognize an object previously presented in terms of the time animals spent exploring a familiar object during a 5-min observation period. The discrimination index was calculated according to the following Equation (2):DI = [(T*_N_* − T*_F_*)/(T*_N_* + T*_F_*)] × 100.(2)
T*_N_* and T*_F_* are the times spent exploring new and familiar objects, respectively, during the 5-min observation period.

#### 4.4.3. Morris Water Maze (MWM) Test

The MWM test was performed to elucidate the spatial reference memory of animals as previously described [10] using a 1.1-m-diameter circular pool. Mice received a block of four trials daily during a 5-day period of training sessions. In each trial, the mouse was placed into the pool at one of 4 start positions 90° apart around the edge of the pool by facing the wall of the tank and allowing the mouse to swim to the hidden transparent platform. This platform was submerged 1.5 cm below the surface of the water in the center of one quadrant (Q1) and, thus, was invisible. The platform position remained stable during the training session. Animals that failed to locate the platform within 60 s were manually guided to the platform by an experimenter. The mouse was allowed to remain on the platform for 10 s before being removed to an opaque high-sided plastic chamber for 60 s. The next trial was then performed. In each trial, the latency to reach the platform (escape latency) and the distance covered were recorded via video capture. After completion of the four trials, mice were kept warm for 1 h and then returned to their home cage. Data for each day were averaged over the 4 trials before being used for result interpretation. In the spatial probe trial, the platform was removed from the pool and a single 60-s probe trial was run to evaluate how well the mice had learned and remembered the exact location of the platform. The times spent in the target quadrant (Q1) and other quadrants (Q2–Q4) of the pool were recorded and compared among the groups.

### 4.5. Measurement of Serum 17β-Estradiol Levels

Twenty hours after completing behavioral experiments, cardiac puncture was performed under nembutal (60 mg/kg, i.e.) anesthesia to obtain an approximately 1-mL blood sample from each mouse. Whole blood was centrifuged at 3000 rpm at 4 °C for 15 min to isolate serum. Supernatants were frozen at −80 °C until biochemical assessments. The serum level of 17β-estradiol was measured by an electrochemiluminescence immunoassay according to the manufacturer’s instructions (ECLIA, Roche e411 immunoassay analyzer).

### 4.6. Dissection of Uterus and Brain Tissues

Animals were sacrificed by decapitation. Their uterus and hippocampus were dissected out and kept at −80 °C until used. The uterus weight and volume from each animal group was measured.

### 4.7. Oxidative Stress Measurement

#### 4.7.1. Lipid Oxidative Damage

Lipid peroxidation in the serum and hippocampus homogenate was measured as previously described [10]. Briefly, the hippocampus was homogenized in 10 vol. of ice-cold phosphate buffer (5 mM, pH 7.4) using a Potter-Elvehjem homogenizer with a Teflon pestle. Serum samples were diluted 4 times with phosphate buffer. Serum or homogenates were mixed with the same volume of 10% (*w*/*v*) trichloroacetic acid and then centrifuged at 8000× *g* at 4 °C for 10 min. The supernatant was incubated with 0.8% (*w*/*v*) 2-thiobarbituric acid at 100 °C for 15 min. After a cooling period, the content of thiobarbituric acid reactive substances (TBARS), an index of lipid peroxidation, was spectrophotometrically measured at 532 nm using malondialdehyde (MDA) as a standard. The amount of TBARS was expressed as nmol MDA/mg protein. The protein contents of serum and hippocampus homogenates were measured by the Bradford method [34].

#### 4.7.2. Assessment of Enzymatic Antioxidant Defenses

The hippocampus was homogenized in cold phosphate buffer (5 mM, pH 7.4) to obtain 20% homogenates. Serum samples were diluted in phosphate buffer to obtain 25% diluted serum. SOD activity was evaluated using a commercially available kit (#19160, Sigma-Aldrich, St. Louis, MO, USA). The method was based on the formation of superoxide radicals produced by xanthine and xanthine oxidase, which consequently reacts with nitroblue tetrazolium (NBT) to form formazan dye. SOD activity was then measured at 560 nm by the degree of inhibition of this reaction. Activity was expressed as U/mg protein. The activity of catalase (CAT) was analyzed using the catalase assay kit (#CAT 100, Sigma-Aldrich, St. Louis, MO, USA) according to the manufacturer’s instruction. One unit of CAT activity is defined as the amount of enzyme catalyzing the degradation of 1 μmole of H_2_O_2_ to oxygen and water per min at 37 °C [34].

### 4.8. Quantitative Real-Time PCR (QPCR)

Mouse PI3K, IL-1β, IL-6 and TNF-α mRNA expressions in frontal cortex were quantified by real-time PCR. Total RNA was extracted from the tissues with TRIzol^®^ (Thermo Fisher Scientific Inc., San Jose, CA, USA) according to the manufacturer’s instructions. First-strand cDNA is synthesized with oligo (dT) primers and SuperScript III reverse transcriptase (Thermo Fisher Scientific Inc., San Jose, CA, USA). QPCR is conducted using SsoAdvanced^TM^ Universal SYBR^®^ Green Supermix (Biorad, Hercules, CA, USA). The following primers were synthesized by Macrogen (Seoul, South Korea): Amplification was carried out using gene-specific PCR primer sets as follow: (1)-β-actin: 5’-AAC GGT CTC ACG TCA GTG TA-3’ (sense) and 5’-GTG ACA GCA TTG CTT CTG TG-3’ (antisense); (2)-PI3K: -5’-GTG TCA GCG CTC TCC GCC 3’ (sense) and 5’ CTG ATA ATT GAT GTA TGG 3’ (antisense); (3)-IL-1β:5’-GAC AGC AAA GTG ATA GGC C-3’ (sense) and 5’-CGT CGG CAA TGT ATG TGT TGG-3’ (antisense); (4)-IL-6: 5’-CTT CCA TCC AGT TGC CTT CTT G-3’ (sense) and 5’-AAT TAA-3’ (antisense); (5)-TNF-α:5’-GCC TCT TCT CAT TCC TGC TTG-3’ (sense) and 5’-CTG ATG AGA GGG AGG CCA TT-3’(antisense). β-Actin mRNA is used as a control. Fold difference relative expressions are calculated.

### 4.9. Statistical Analysis

Data are expressed as means ± SEM and were analyzed by the paired Student’s t-test for the Sham and OVX groups or a one-way ANOVA followed by test for multiple comparisons among different groups. Differences of *p* < 0.05 were considered to be significant. Analyses were conducted using SigmaStat^®^ ver. (SYSTAT Software Inc., Richmond, CA, USA).

## 5. Conclusions

In conclusion, the present results suggest that PM extract exerts beneficial effects on cognitive deficits attributable to menopause/estrogen depletion mainly via the phytoestrogenic, antioxidant activities. Moreover, this study extends affirmative evidence, for the first time, that PM exhibits the neuroinflammatory effect by regulation of proinflammatory cytokines involved in AD such as IL-1β, IL-6 and TNF-α, and estrogen-mediated gene, PI3K mRNA in the hippocampus. The efficacious action of PM as a result of phytoestrogens, such as miroestrol, deoxymiroestrol, isomiroestrol, puerarin, daidzin, genistin, daidzein, genistein, and kwakurin, were constituents in the extract.

## Figures and Tables

**Figure 1 molecules-26-03442-f001:**
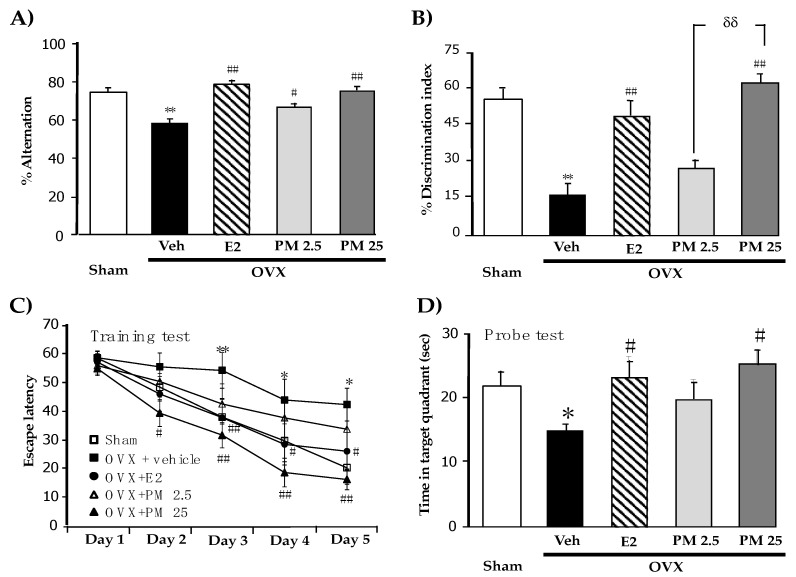
Effects of PM extract and estrogen on OVX-induced cognitive impairments. (**A**) represent the result from Y-maze test. (**B**) represent the result from NORT. (**C**) and **(D**) represent the result from MWM test. Values given were the mean ± S.E.M. (n = 10–15). * *p* < 0.05, ** *p* < 0.001 vs. sham-operated group, ^#^
*p* < 0.05, ^##^
*p* < 0.001 vs. the OVX group and **^δδ^**
*p* < 0.001 vs. the PM extract (post-hoc Tukey test).

**Figure 2 molecules-26-03442-f002:**
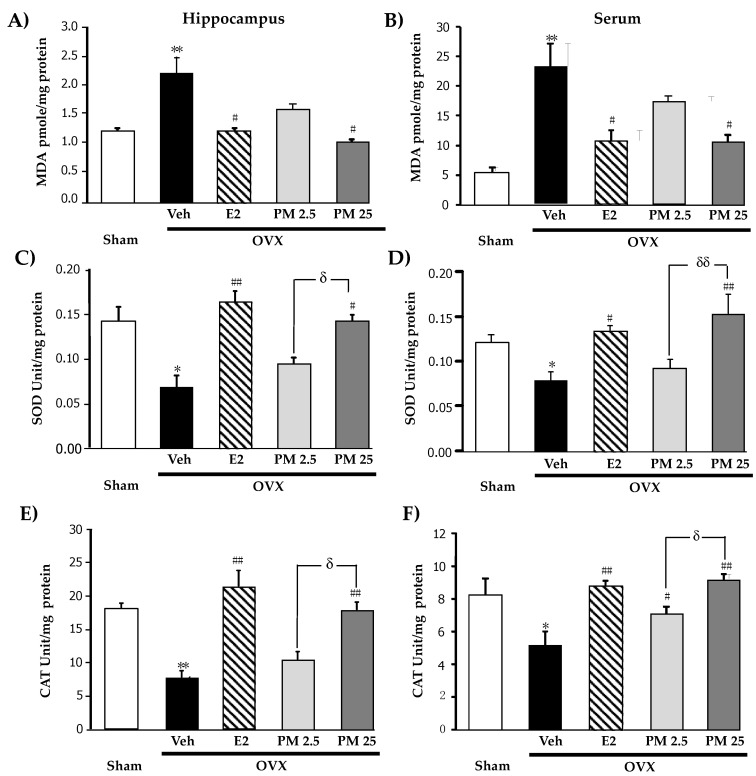
Effects of PM extract and estrogen on oxidative damage and antioxidant enzyme activities. (**A**,**B**) were the effects of PM extract and estrogen on oxidative damage in hippocampus and serum, respectively. (**C**,**D**) were the effects of PM extract and estrogen on SOD enzyme activity in hippocampus and serum, respectively. (**E**,**F**) were the effects of PM extract and estrogen on CAT enzyme activity in hippocampus and serum, respectively. The values given were the mean ± SEM (n = 5–6). * *p* < 0.05, ** *p* < 0.001 vs. sham-operated group, ^#^
*p* < 0.05, ^##^
*p* < 0.001 vs. the OVX group, **^δ^**
*p* < 0.05 and **^δδ^**
*p* < 0.001 *p* < 0.05 vs. the PM extract (post-hoc Tukey test).

**Figure 3 molecules-26-03442-f003:**
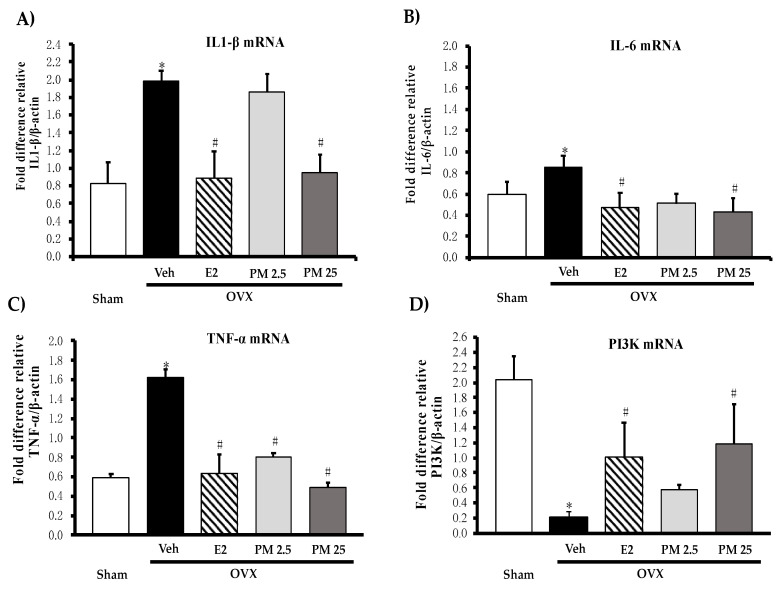
Effects of PM extract and 17β-estradiol on estrogen-mediated gene on proinflammatory cytokines (**A–C**) and estrogen-mediated gene (**D**) in hippocampus. Values given were the mean ± SEM (n = 5–6). * *p* < 0.05 vs. sham-operated group and ^#^
*p* < 0.05 vs. the OVX group (post-hoc Tukey test).

**Figure 4 molecules-26-03442-f004:**
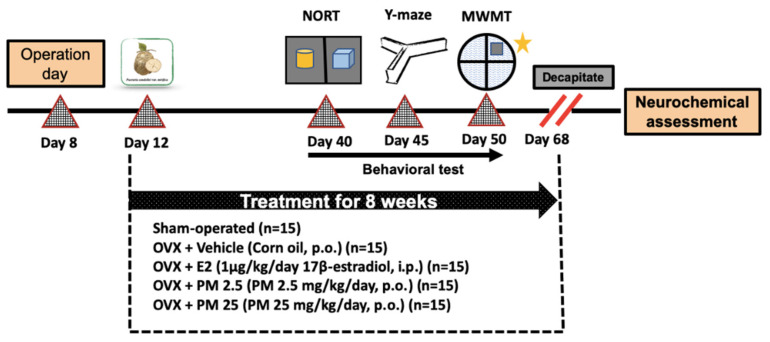
A schematic representation of the experiment framework.

**Table 1 molecules-26-03442-t001:** The uterus weight and volume and serum 17β-estradiol levels after the PM treatment.

Treatment	Uterus		Serum E2 (pg/mL)
	Weight (g)	Volume (cm^3^)	
Sham	0.227 ± 0.014	0.153 ± 0.022	24.63 ± 2.27
OVX + vehicle	0.069 ± 0.008 *	0.054 ± 0.009 *	13.03 ± 2.38 *
OVX + E2 (1 μg/kg/day)	0.277 ± 0.028 ^#^	0.135 ± 0.010 ^#^	32.88 ± 4.99 ^##^
OVX + PM (2.5 mg/kg/day)	0.199 ± 0.009 ^#^	0.163 ± 0.006 ^#^	16.10 ± 2.51
OVX + PM (25 mg/kg/day)	0.226 ± 0.013 ^#^	0.179 ± 0.016 ^#^	23.74 ± 2.95 *

Values given were the mean ± SEM (n = 10–12), ** p* < 0.05 vs. sham-operated group, ^#^
*p* < 0.05, ^##^
*p* < 0.001 vs. the OVX group (post-hoc Tukey test).

## Data Availability

Not applicable.

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
