# Peer review of "Effects of Pueraria candollei var mirifica (Airy Shaw and Suvat.) Niyomdham on Ovariectomy-Induced Cognitive Impairment and Oxidative Stress in the Mouse Brain"

_molecules, 2021, doi:10.3390/molecules26113442_

Round 1

Reviewer 1 Report

The submitted manuscript titled:  Effects of Pueraria candollei var. mirifica (Airy Shaw & Suvat.) Niyomdham on Ovariectomy-Induced Cognitive Impairment and Oxidative Stress in the Mouse Brain is very interesting for readers. The Authors showed that PM extract can down-regulate expression of pro-inflammatory cytokines, up-regulate expression of PI3K and cause a decrease in MDA level in brain. Although the conclusions reached are of great interest, some aspects of the study should be reviewed.

  1. Abstract – page 1, line 29-30: “… IL-1β, IL-6 and TNF-α proinflammatory cytokines” it should be rewritten
  2. Introduction – the explanation of abbreviations: APP and Aβ should be introduced after their first appearance in the text (page 2, lines 60-61)
  3. Results:
  • on page 3 (line 116), page 4 (line 125, 131 and 134-135), Authors described the results presented on Figure 1, but in the text they referred to Figure 2.
  • Page 5 (lines 151-159) – the lack of the reference to presentation of the results (e.g. Figure 2?). This mistake applies also page 6, lines 164-171.
  • Figure 2 and 3 – it should be divided into subsection A, B, C, D… like Figure 1
  • Under Figure and Table 1 – the lack of explanation of statistically significance symbols

In manuscript there are few typos e.g. title of Figure 4, page 10 line 278: “Fifty-five 5-week-old female mice”

Author Response

Please see the attachment (response to reviewer 1#)

Reviewer 2 Report

Having read the manuscript entitled “Effects of Pueraria candollei var. mirifica (Airy Shaw & Suvat.)

Niyomdham on ovariectomy-induced cognitive impairment and oxidative stress in the mouse brain” I have to admit that the subject is interesting, the manuscript is well-written, and the obtained data are clearly presented. However, there are several issues that have to be changed before publication:

*Introduction

In the Introduction section Authors little mentioned about the presented studies. Therefore, more information about the study should be added,  i.e., its importance, tested parameters, study hypotheses, its significance for human population.

*Results

The Result section lacks the outcomes of the statistical analyses (p, F).

Page 4, line 123: according to the data presented in Fig. 1B, only the higher dose of PM was active

Page 4, line 125: it should be Figure 1B

Page 4, line 131: it should be Figure 1C and D

Page 4, line 133: according to the data presented in Fig. 1D, only the higher dose of PM was active

Page 4, line 134: it should be Figure 1C

Page 5, line 157: according to the Material and Methods section PM was given i.p.

Section 2.3. lacks reference to Figure 2.

Section 2.4. lacks reference to Figure 3

Page 6, line 169: according to Figure 3, PM at a dose of 2.5 was not active in relation to the changes in Il-1b, Il-6, and PI3K mRNA.

*Discussion

The Discussion section lacks a short paragraph concerning limitations of the study.

* Animals

Mice strain should be specified.

How many mice were kept in 1 cage?

*Surgical procedure

On what basis did the Authors select the tested doses of E2 and PM? And on what basis did the Authors choose the schedule of the experiments?

Why the vehicle was given p.o. whereas the tested substances were given i.p.?

How E2/PM dilutions were prepared? Were they solutions or suspensions?

*Figures/table

Figure numbers are wrong.

Figure 3 should have the same appearance as Fig. 1 and Fig. 2.

The enclosed figures lack figure captions. Similarly, the enclosed Table lacks a proper description. Data presented in figures and tables should be understandable without referring to the main text. Therefore, figure captions/table descriptions should give information about the tested agents, their doses, treatment schedule, number of animals per group, statistical test etc. Moreover, significant differences between the tested groups indicated in the Figures and the Table should be explained.

Author Response

Please see the attachment (see the response to reviewer 2#).

Round 2

Reviewer 1 Report

I recommend this manuscript for publication.

Reviewer 2 Report

All of my suggestions have been addressed.